# The Performance of a Deep Learning-Based Automatic Measurement Model for Measuring the Cardiothoracic Ratio on Chest Radiographs

**DOI:** 10.3390/bioengineering10091077

**Published:** 2023-09-12

**Authors:** Donguk Kim, Jong Hyuk Lee, Myoung-jin Jang, Jongsoo Park, Wonju Hong, Chan Su Lee, Si Yeong Yang, Chang Min Park

**Affiliations:** 1Institute of Medical and Biological Engineering, Medical Research Center, Seoul National University, 101, Daehak-ro, Jongno-gu, Seoul 03080, Republic of Korea; drinkuranium@snu.ac.kr; 2Department of Radiology, Seoul National University College of Medicine, Seoul National University Hospital, 101, Daehak-ro, Jongno-gu, Seoul 03080, Republic of Korea; 3Medical Research Collaborating Center, Seoul National University Hospital, 101, Daehak-ro, Jongno-gu, Seoul 03080, Republic of Korea; 4Department of Radiology, College of Medicine, Yeungnam University 170, Hyeonchung-ro, Nam-gu, Daegu 42415, Republic of Korea; 5Department of Radiology, Hallym University Sacred Heart Hospital, Anyang-si, Gyeonggi-do 14068, Republic of Korea; 6Center for Artificial Intelligence in Medicine and Imaging, HealthHub Co. Ltd., 623, Gangnam-daero, Seocho-gu, Seoul 06524, Republic of Korea; 7Institute of Radiation Medicine, Seoul National University Medical Research Center, 101, Daehak-ro, Jongno-gu, Seoul 03080, Republic of Korea

**Keywords:** chest radiographs, cardiothoracic ratio, deep learning, artificial intelligence, validation

## Abstract

Objective: Prior studies on models based on deep learning (DL) and measuring the cardiothoracic ratio (CTR) on chest radiographs have lacked rigorous agreement analyses with radiologists or reader tests. We validated the performance of a commercially available DL-based CTR measurement model with various thoracic pathologies, and performed agreement analyses with thoracic radiologists and reader tests using a probabilistic-based reference. Materials and Methods: This study included 160 posteroanterior view chest radiographs (no lung or pleural abnormalities, pneumothorax, pleural effusion, consolidation, and *n* = 40 in each category) to externally test a DL-based CTR measurement model. To assess the agreement between the model and experts, intraclass or interclass correlation coefficients (ICCs) were compared between the model and two thoracic radiologists. In the reader tests with a probabilistic-based reference standard (Dawid–Skene consensus), we compared diagnostic measures—including sensitivity and negative predictive value (NPV)—for cardiomegaly between the model and five other radiologists using the non-inferiority test. Results: For the 160 chest radiographs, the model measured a median CTR of 0.521 (interquartile range, 0.446–0.59) and a mean CTR of 0.522 ± 0.095. The ICC between the two thoracic radiologists and between the model and two thoracic radiologists was not significantly different (0.972 versus 0.959, *p* = 0.192), even across various pathologies (all *p*-values > 0.05). The model showed non-inferior diagnostic performance, including sensitivity (96.3% versus 97.8%) and NPV (95.6% versus 97.4%) (*p* < *0*.001 in both), compared with the radiologists for all 160 chest radiographs. However, it showed inferior sensitivity in chest radiographs with consolidation (95.5% versus 99.9%; *p* = 0.082) and NPV in chest radiographs with pleural effusion (92.9% versus 94.6%; *p* = 0.079) and consolidation (94.1% versus 98.7%; *p* = 0.173). Conclusion: While the sensitivity and NPV of this model for diagnosing cardiomegaly in chest radiographs with consolidation or pleural effusion were not as high as those of the radiologists, it demonstrated good agreement with the thoracic radiologists in measuring the CTR across various pathologies.

## 1. Introduction

Chest radiographs are the most common and basic diagnostic examination for cardiothoracic and pulmonary diseases, accounting for 40% of all radiologic examinations [1,2]. The cardiothoracic ratio (CTR) measured on chest radiographs, defined as the ratio of the greatest transverse dimension of the heart to the greatest transverse dimension of the thoracic cavity, is a simple but critical parameter for assessing cardiomegaly [3,4]. Since various heart diseases are accompanied by cardiomegaly (e.g., hypertension, coronary artery disease, cardiac valve disease, and pulmonary hypertension), accurate CTR measurements can help initiate diagnostic workups for patients’ underlying heart diseases, potentially leading to an improvement in patients’ prognosis [5,6,7,8,9]. Despite the clinical importance of the CTR, because of the vast number of chest radiographs in the increasingly modern medical workload, measuring the CTR on all chest radiographs is considered a time-consuming and redundant step [2,10]. In addition, since human measurements of the CTR (e.g., by radiologists or cardiologists) are considered the gold standard, there are issues with intra-observer and inter-observer variability in terms of accurately measuring the CTR [3].

Deep learning (DL) has recently been applied to various medical tasks and achieved a superior or comparable diagnostic performance to experts [11]. Indeed, prior studies reported that automatic CTR measurement using DL, specifically U-Net, could provide accurate CTR measurements on chest radiographs [5,12,13,14,15,16]. However, those studies used reference standards constructed with only one radiologist or a consensus reading of two or three radiologists, did not evaluate model performance in the setting of various thoracic pathologies, or did not conduct reader tests with multiple readers; these methodological aspects of previous studies can limit the applicability of DL in measuring the CTR in real-world clinical settings [5,12,13,14,15,16]. Therefore, this study aimed to validate the performance of a commercially available DL-based CTR measurement model with various thoracic pathologies, and perform agreement analyses with thoracic radiologists and then reader tests using a probabilistic-based reference standard.

## 2. Materials and Methods

### 2.1. DL-Based Model Measuring the CTR on Chest Radiographs

A commercially available DL-based model measuring the CTR on chest radiographs (CTR-AI, version 1.0, HealthHub) was used in this study. Specifically, the model extracts boundaries of the lungs and heart on chest radiographs (posteroanterior view). The architecture of this system is presented in Figure 1. Using standard U-Net architecture, two DL algorithms segment the lungs and heart. These DL algorithms were trained with 633 chest radiographs and internally validated with 160 radiographs from the Japanese Society of Radiological Technology, the Montgomery public dataset, and an in-house dataset. Self-attention modules, consisting of a channel and spatial attention blocks, were added to improve the ability to represent disparate features [14,17,18]. The channel attention block extracts the inter-channel connections of the input feature map, while the spatial channel block encodes the relative importance of each spatial location of the input feature map. These attention blocks can be located in the U-Net architecture at any place and in any number. In the best-performance experiments, the attention modules were applied in the first and second places in both directions of the U-Net architecture. The detailed architecture of the segmentation model is shown in Figure 1. After segmenting the lungs and heart, the DL-based model calculates the maximum horizontal distances for each segmented area using an image-processing algorithm to output the CTR value.

### 2.2. Study Sample

To validate the DL model in the setting of various pathologies, we collected posteroanterior chest radiographs with the following findings: no lung or pleural abnormalities (i.e., chest radiographs without any lung parenchymal or pleural abnormalities) (*n* = 40), pneumothorax (*n* = 40), pleural effusion (*n* = 40), and lung consolidation (*n* = 40). We randomly collected a study sample among chest radiographs taken between July 2018 and June 2021.

### 2.3. Measurement of the CTR by Thoracic Radiologists for Agreement Analyses

To assess the agreement between the DL model and human experts in measuring the CTR on chest radiographs, two thoracic radiologists measured the CTR of the 160 study sample radiographs. They independently measured the CTR twice at a 1-month interval (washout period). That is, four datasets for testing were obtained (two datasets from each of the two thoracic radiologists). When measuring the CTR, they were instructed to measure the maximum left heart diameter (MLD), the maximum right heart diameter (MRD), and the greatest transverse dimension of the thoracic cavity (GT). Then, the CTR was calculated as (MLD + MRD)/GT (Figure 2).

### 2.4. Reader Tests

To compare the diagnostic performance of the DL model for diagnosing cardiomegaly to that of board-certified radiologists, five board-certified radiologists, who did not participate in the agreement analyses, independently measured the CTR of the test datasets in the same way as in the agreement analyses.

Since cardiomegaly is usually defined as a CTR value of more than 0.50 [4], we applied this cut-off value in the reader tests. Since the gold standard of measuring CTR is by human experts [4], we constructed the reference standard using the four measurement results obtained in the agreement analysis. To construct a reference standard for diagnosing cardiomegaly, the Dawid–Skene consensus method was used as a robust way for determining ground truth from various labeling data [19,20]. Specifically, we first categorized the four datasets from the two thoracic radiologists as a categorical variable (i.e., the presence or absence of cardiomegaly) using a cut-off value of 0.50. Then, a probabilistic generative model (Dawid–Skene consensus method) was used to fuse the labels from multiple annotation results by weighting reliable factors [19,20]. In addition, we set a cut-off value for the cardiomegaly as 0.55, since several prior studies referred to a CTR value of 0.55 for significant cardiomegaly [5,21,22].

As a sensitivity analysis, we set the reference standard with the median values of the four datasets per case. 

### 2.5. Statistical Analysis

To calculate the sample size of the test dataset, the Bland–Altman plot was used with an expected mean of differences of 0.00175, an expected standard deviation of differences of 0.04108, and a maximum allowed difference between methods of 0.15. Type I error (alpha) and type II error (beta) were both set at 0.05. The resultant minimum sample size was 23.

Continuous variables are presented as mean with standard deviation (SD) or median with interquartile range (IQR). To evaluate the agreement in measurements of the CTR between thoracic radiologists (four datasets) and the DL-based model, we used the following methods: (a) Bland–Altman plots and mean absolute or relative differences with their limits of agreement (LOAs) [23,24]; (b) mean absolute error (MAE) and root mean square error (RMSE) between thoracic radiologists and the DL-based model; (c) intraclass or interclass correlation coefficients with a comparison of interclass correlation coefficients (ICCs) between the thoracic radiologists and between the thoracic radiologists and the DL-based model [25]. The *p* value was calculated from the empirical distribution from 1000 bootstrap samples.
$$\mathrm{MAE}=\frac{1}{n}\sum_{t=1}^{n}\left|y_{i}-x_{i}\right|,\;\mathrm{RMSE}=\sqrt{\frac{1}{n}\sum_{t=1}^{n}{\left(y_{i}-x_{i}\right)}^{2}}$$
where y_i_ (e.g., DL model) and x_i_ (e.g., the thoracic radiologists) represent two measurement values.

In the reader tests, diagnostic measures, including sensitivity, specificity, positive predictive value (PPV), negative predictive value (NPV), and accuracy of the DL-based model and five board-certified radiologists were calculated and compared using the non-inferiority test with a non-inferiority limit of 10%.

Subgroup analyses were performed in chest radiographs with no lung or pleural abnormalities, pneumothorax, pleural effusion, and lung consolidation. 

All statistical analyses were performed using R version 4.1.0 (R Project for Statistical Computing). A *p* value < 0.05 was considered to indicate statistical significance, but a cut-off of a *p* value of 0.025 was used in the one-sided non-inferiority test.

## 3. Results

### 3.1. Study Sample and CTR

In total, 160 individuals (80 men and 80 women; mean age 53.7 ± 19.1 years) and their chest radiographs were included in this study. The median and mean CTR measured by the DL-based model were 0.521 (interquartile range [IQR], 0.446–0.59) and 0.522 ± 0.095, respectively. In the first session, two thoracic radiologists measured the median CTR as 0.501 (IQR, 0.448–0.566) and 0.511 (IQR, 0.447–0.578) and the mean CTR as 0.51 ± 0.088 and 0.518 ± 0.092, respectively. In the second session, they measured the median CTR as 0.499 (IQR, 0.439–0.563) and 0.509 (IQR, 0.449–0.582) and the mean CTR as 0.509 ± 0.088 and 0.52 ± 0.092, respectively.

### 3.2. Agreement Analyses

The mean absolute difference of CTR between the DL-based model and the two thoracic radiologists was 0.0074 with a lower 95% LOA of −0.0457 (95% confidence interval [CI]: −0.052, −0.0403) and an upper 95% LOA of 0.0605 (95% CI: 0.0551, 0.0668). The mean relative difference between the DL model and the two thoracic radiologists was 1.3%, with a lower 95% LOA of −9.33% (95% CI: −10.6%, −8.23%) and an upper 95% LOA of 11.92% (95% CI: 10.82%, 13.2%) (Figure 3 and Table 1). In the subgroup analysis, chest radiographs with pleural effusion showed a mean absolute difference of 0.0153, with a lower 95% LOA of −0.0418 (95% CI: −0.0572, −0.0305) and an upper 95% LOA of 0.0724 (95% CI: 0.0611, 0.0877), and a mean relative difference of 3.12% with a lower 95% LOA of −8.68% (95% CI: −11.93%, −6.29%) and an upper 95% LOA of 14.91% (95% CI: 12.52%, 18.16%) (Figure 4 and Appendix A). In other words, the difference in CTR between the DL-based model and the two thoracic radiologists within these ranges were allowed with 95% confidence.

The MAE and RMSE between the DL-based model and the two thoracic radiologists in two sessions were 0.019 and 0.028, respectively. In the subgroup analyses, the MAE and RMSE in chest radiographs without any lung or pleural abnormality, chest radiographs with pneumothorax, chest radiographs with pleural effusion, and chest radiographs with consolidation were 0.013 and 0.021, 0.013 and 0.02, 0.025 and 0.033, and 0.024 and 0.035, respectively (Appendix A).

The intra-observer correlation coefficients of the thoracic radiologists were 0.988 (95% CI: 0.983, 0.991) and 0.977 (95% CI: 0.968, 0.983), respectively. The inter-observer correlation coefficient was calculated as 0.972 (95% CI: 0.953, 0.982) between the two thoracic radiologists and 0.959 (95% CI: 0.945, 0.971) between the DL-based model and the two thoracic radiologists, and these agreements were not significantly different (*p* = 0.192). In the subgroup analyses, there were no significant differences in agreement between the two thoracic radiologists and between the model and the two thoracic radiologists (*p* values > 0.05) (Table 2).

### 3.3. Reader Tests

With the reference standard for cardiomegaly constructed from the Dawid–Skene consensus method, the DL-based model showed a sensitivity of 96.3% (95% CI: 89.7%, 99.2%), specificity of 83.3% (95% CI: 73.2%, 90.8%), PPV of 85.9% (95% CI: 77%, 92.3%), NPV of 95.6% (95% CI: 87.6%, 99.1%), and accuracy of 90% (95% CI: 84.3%, 94.2%) for diagnosing cardiomegaly. The five board-certified radiologists had a sensitivity of 97.8% (95% CI: 95.1%, 99%), specificity of 85.1% (95% CI: 78.8%, 89.8%), PPV of 87.4% (95% CI: 81.1%, 91.8%), NPV of 97.4% (95% CI: 93.9%, 98.9%), and accuracy of 91.6% (95% CI: 88.1%, 94.2%). The DL-based model showed non-inferiority to the board-certified radiologists in all diagnostic measures (*p* values < 0.025) (Table 3). In the subgroup analyses, the DL-based model produced non-inferior results of sensitivity, PPV, NPV, and accuracy compared to the radiologists in chest radiographs without any lung or pleural abnormality. The model showed comparable sensitivity, NPV, and accuracy for chest radiographs with pneumothorax, and only comparable sensitivity for chest radiographs with pleural effusion. Finally, the model exhibited non-inferiority in specificity, PPV, and accuracy for chest radiographs with consolidation (Figure 5). The results of reader tests with the definition of cardiomegaly as CTR of 0.55 are described in Appendix A.

When the median CTR values were used as the reference standard, the DL-based model showed a sensitivity of 97.6% (95% CI: 91.7%, 99.7%), specificity of 86.8% (95% CI: 77.1%, 93.5%), PPV of 89.1% (95% CI: 80.9%, 94.7%), NPV of 97.1% (95% CI: 89.8%, 99.6%), and accuracy of 92.5% (95% CI: 87.3%, 96.1%). The five board-certified radiologists had a sensitivity of 97.9% (95% CI: 95.8%, 98.9%), specificity of 87.4% (95% CI: 81.9%, 91.4%), PPV of 89.5% (95% CI: 84%, 93.3%), NPV of 97.4% (95% CI: 94.5%, 98.7%), and accuracy of 92.9% (95% CI: 89.9%, 95%). The DL-based model achieved equivalent diagnostic measures to those of the board-certified radiologists (*p* values < 0.025) (Appendix A).

## 4. Discussion

In this study, we validated the performance of a commercially available DL-based CTR measurement model by assessing its agreement with thoracic radiologists, and then performed reader tests with five radiologists in various thoracic pathologies. The mean absolute and relative differences between the DL-based model and thoracic radiologists were 0.0074 and 1.3%, and their error ranges were from −0.0457 to 0.0605 and from −9.33% to 11.92%, respectively. The MAE and RMSE between the model and the two thoracic radiologists were 0.019 and 0.028, respectively. The ICC between the model and the thoracic radiologists was 0.959, comparable to that between the two thoracic radiologists (ICC = 0.972; *p* = 0.192). Finally, the DL-based model had comparable sensitivity, specificity, PPV, NPV, and accuracy for diagnosing cardiomegaly compared with five board-certified radiologists using both the reference standard constructed by Dawid–Skene consensus (sensitivity, 96.3% versus 97.8%; specificity, 83.3% versus 85.1%; PPV, 85.9% versus 87.4%; NPV, 95.6% versus 97.4%; and accuracy, 90.0% versus 91.6%; all *p* < 0.025) and the median CTR determined by the thoracic radiologists (sensitivity, 97.6% versus 97.9%; specificity, 86.8% versus 87.4%; PPV, 89.1% versus 89.5%; NPV, 97.1% versus 97.4%; and accuracy, 92.5% versus 92.9%; all *p* < 0.025).

We used the Dawid–Skene consensus method to construct reference standards. This is a statistical method for determining the ground truth based on a probabilistic generative model for fusing the labels from multiple voters in a coherent manner [19]. Specifically, the Dawid–Skene model generates ground truth from multiple voters by discounting unreliable factors’ contributions while compensating them with reliable factors for model prediction [19,26]. In contrast, prior studies set the reference standards for measuring the CTR using only one radiologist or consensus readings of two or three radiologists, which are prone to inter- or intra-observer variability [12,14,16].

Another point to consider in measuring CTR is that various lung or pleural pathologies can exist in real-world clinical settings (e.g., pneumothorax, pleural effusion, and lung opacity). Although a prior study included these pathologies in their study sample, they reported only the measurement results of their DL-based model in these settings without reader tests, limiting the clinical validity of the model [14]. In contrast, we validated our DL-based model in these pathologic settings through agreement analyses with thoracic radiologists and reader tests with board-certified radiologists. The model showed agreement with thoracic radiologists that was equivalent to those between the two thoracic radiologists in all pathologic settings. Since DL-based models have the potential to be used as screening or triaging tools for cardiomegaly in a real-world clinical setting [5,6,7,8,9], sensitivity and NPV are key diagnostic factors in such a setting [27,28]. In the reader tests, the DL-based model achieved diagnostic measures of sensitivity and NPV that were comparable to the five board-certified radiologists for chest radiographs without any lung or pleural abnormality, as well as for chest radiographs with pneumothorax. In contrast, the model achieved an inferior NPV for chest radiographs with pleural effusion, and inferior sensitivity and NPV for chest radiographs with consolidation. These results are in line with a prior study, according to which a DL-based model exhibited low CTR measurement performance in chest radiographs with abnormal findings obscuring the margin of the thoracic cage (e.g., pleural effusion) and heart border (e.g., pneumonia) [14].

Two limitations should be noted in this study. First, we only assessed the measurement performance of the DL-based model for the CTR without evaluating the added value of the model compared to the human radiologists (e.g., improvement in diagnostic performance, reduced measurement time to diagnose cardiomegaly) [29,30]. Second, although we showed a comparable performance of the DL-based model in measuring the CTR and diagnosing cardiomegaly, its clinical applicability in a specific scenario was not investigated. For instance, a prime example would be applying this DL-based model in the emergency department to triage patients who should be seen by a cardiologist first. Further validation studies are warranted.

In conclusion, while the sensitivity and NPV of this DL-based model for diagnosing cardiomegaly in chest radiographs with pleural effusion or consolidation were not as high as those of radiologists, the model demonstrated good agreement with thoracic radiologists in measuring the CTR across various pathologies. 

## Figures and Tables

**Figure 1 bioengineering-10-01077-f001:**
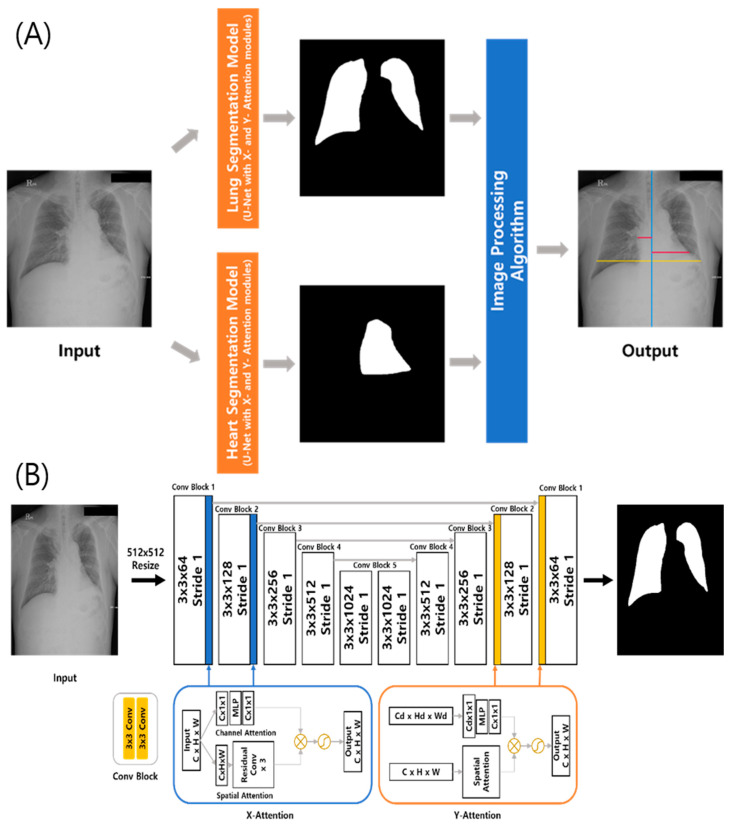
(**A**) Architecture of the computer-aided automatic measurement system of the cardiothoracic ratio (CTR) on chest radiographs. (**B**) Detail of the segmentation model using standard U-Net architecture with specific self-attention modules.

**Figure 2 bioengineering-10-01077-f002:**
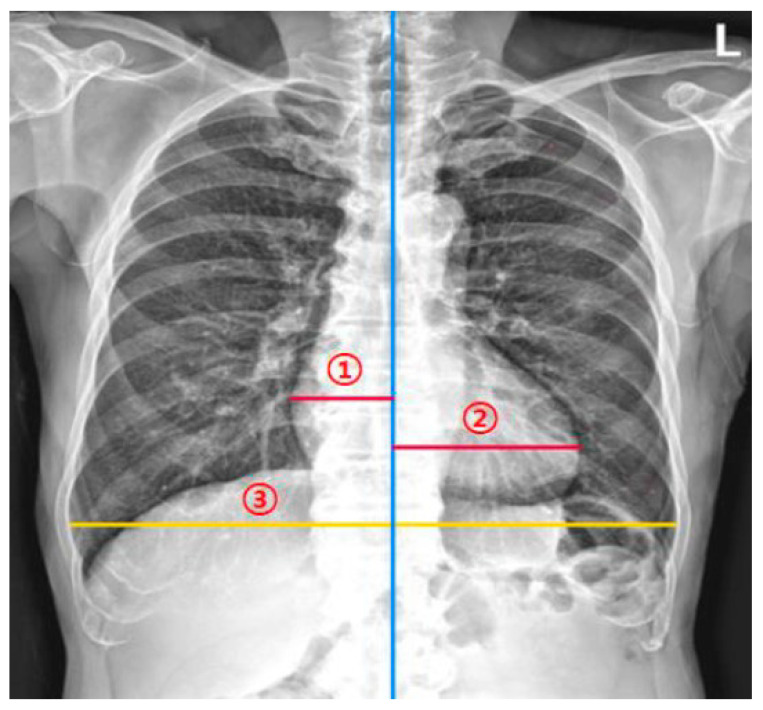
Measurement of the cardiothoracic ratio (CTR). The CTR measurement model and radiologists measured (**1**) the maximum left heart diameter, (**2**) the maximum right heart diameter, and (**3**) the greatest transverse dimension of the thoracic cavity. The CTR is calculated as (maximum left heart diameter + the maximum right heart diameter)/the greatest transverse dimension of the thoracic cavity.

**Figure 3 bioengineering-10-01077-f003:**
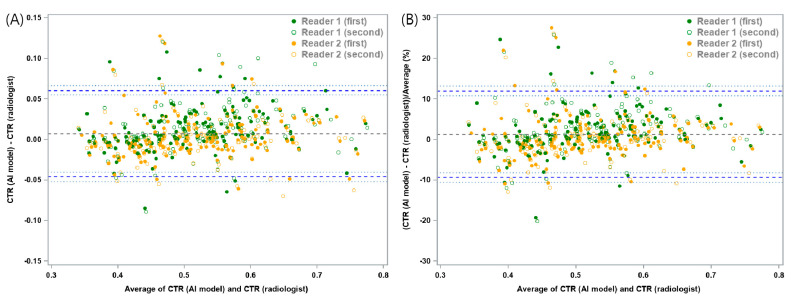
Bland–Altman plots for cardiothoracic ratio (CTR) measurements between the deep learning-based model and thoracic radiologists. (**A**) The mean absolute difference was 0.0074, with a lower 95% limit of agreement (LOA) of−0.0457 (95% confidence interval [CI]: −0.052, −0.0403) and an upper 95% LOA of 0.0605 (95% CI: 0.0551, 0.0668). (**B**) The mean relative difference between the model and the two thoracic radiologists was 1.3%, with a lower 95% LOA of −9.33% (95% CI: −10.6%, −8.23%) and an upper 95% LOA of 11.92% (95% CI: 10.82%, 13.2%). The thick blue dotted line represents the 95% LOA for the differences, while the thin dotted blue line represents a 95% estimate of the LOA.

**Figure 4 bioengineering-10-01077-f004:**
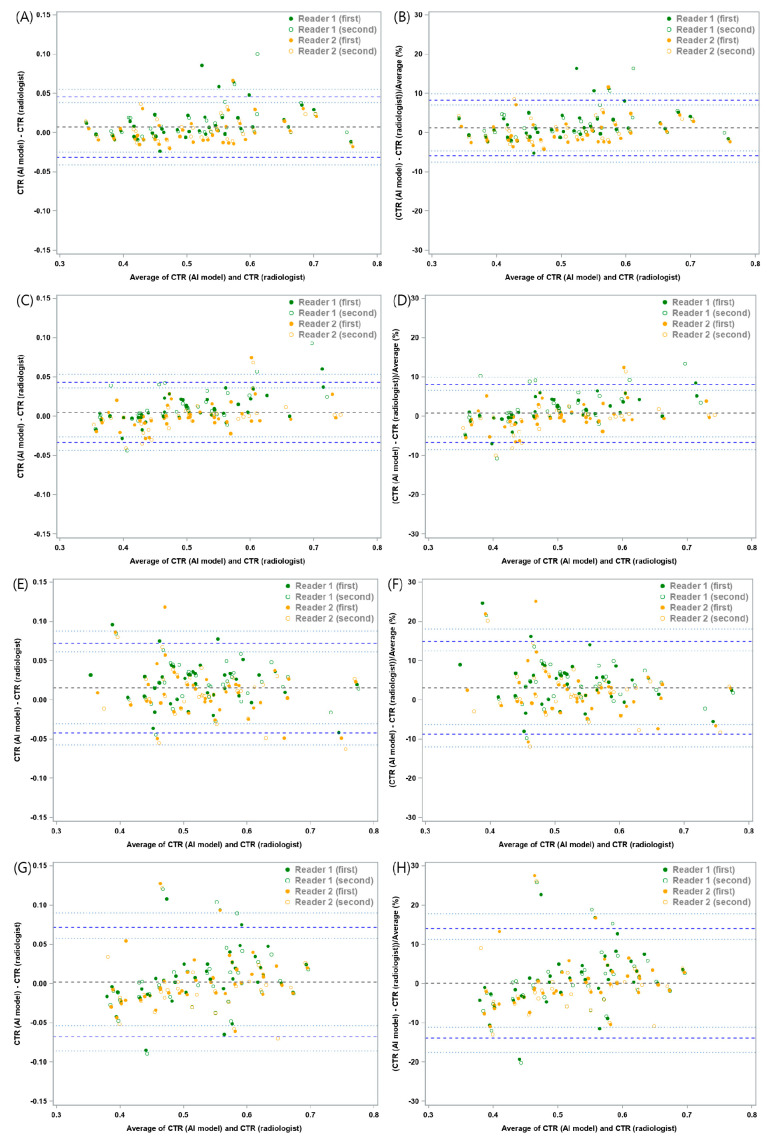
Bland–Altman plots for cardiothoracic ratio (CTR) measurements between the deep learning-based model and thoracic radiologists in various thoracic pathologies. The mean absolute and relative differences in chest radiographs (**A**,**B**) without any lung or pleural abnormality, (**C**,**D**) with pneumothorax, (**E**,**F**) with pleural effusion, and (**G**,**H**) with lung consolidation. Detailed information for these are shown in Table 1.

**Figure 5 bioengineering-10-01077-f005:**
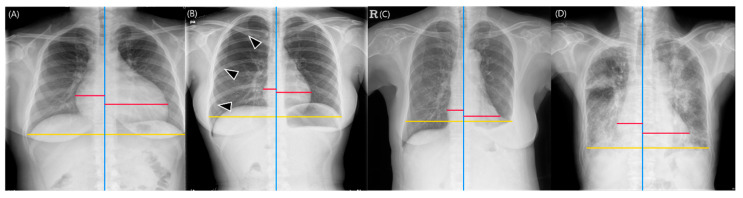
Representative images of cardiothoracic ratio (CTR) measurements obtained from the deep learning-based model. The red lines represent the maximum left and right heart diameters, respectively, the yellow lines indicate the greatest transverse dimension of the thoracic cavity, and the blue lines represent the vertical lines passing through the midpoint of the vertebral bodies. (**A**) Chest radiographs without any lung or pleural abnormality. The deep learning-based model calculated the CTR as 0.588, which was determined as indicative of cardiomegaly. All five board-certified radiologists identified the radiograph as demonstrating cardiomegaly (CTR range: 0.586–0.599). (**B**) Chest radiographs with pneumothorax (arrowheads). The model calculated the CTR as 0.362 (CTR range of five radiologists, 0.356–0.375). (**C**) Chest radiographs with pleural effusion. The model calculated the CTR as 0.5. However, the chest cavity was measured to be shorter than its actual size due to the left pleural effusion (CTR range of five radiologists, 0.433–0.552). (**D**) Chest radiographs with lung consolidation. The CTR measured by the model was 0.593, which was determined as cardiomegaly, and all five board-certified radiologists read this radiograph as having cardiomegaly (CTR range, 0.554–0.595).

**Table 1 bioengineering-10-01077-t001:** Mean differences and mean relative differences between measurements of the cardiothoracic ratio by a deep learning-based model and two thoracic radiologists.

	**Mean Absolute** **Difference**	**Lower LOA (95% CI)**	**Upper LOA (95% CI)**
Study sample (*n* = 160)	0.0074	−0.0457 (−0.052, −0.0403)	0.0605 (0.0551, 0.0668)
Chest radiographs without any lung or pleural abnormality (*n* = 40)	0.0071	−0.0315 (−0.041, −0.0245)	0.0458 (0.0387, 0.0553)
Chest radiographs with pneumothorax (*n* = 40)	0.0051	−0.0334 (−0.0432, −0.0262)	0.0436 (0.0363, 0.0533)
Chest radiographs with pleural effusion (*n* = 40)	0.0153	−0.0418 (−0.0572, −0.0305)	0.0724 (0.0611, 0.0877)
Chest radiographs with consolidation (*n* = 40)	0.0022	−0.0673 (−0.0857, −0.0537)	0.0717 (0.0581, 0.0901)
	**Mean Relative** **Difference (%)**	**Lower LOA (95% CI)**	**Upper LOA (95% CI)**
Study sample (*n* = 160)	1.3	−9.33 (−10.6, −8.23)	11.92 (10.82, 13.2)
Chest radiographs without any lung or pleural abnormality (*n* = 40)	1.22	−5.85 (−7.54, −4.59)	8.29 (7.03, 9.98)
Chest radiographs with pneumothorax (*n* = 40)	0.73	−6.63 (−8.5, −5.24)	8.08 (6.7, 9.96)
Chest radiographs with pleural effusion (*n* = 40)	3.12	−8.68 (−11.93, −6.29)	14.91 (12.52, 18.16)
Chest radiographs with consolidation (*n* = 40)	0.12	−13.84 (−17.58, −11.09)	14.09 (11.34, 17.83)

LOA: limit of agreement; CI: confidence interval.

**Table 2 bioengineering-10-01077-t002:** Intraclass or interclass correlation coefficient (ICC) analysis for measurements of the cardiothoracic ratio on chest radiographs between a deep learning-based model and two thoracic radiologists.

	Agreement	ICC (95% Confidence Interval)	*p*-Value *
Study sample	Intra-observer agreement (thoracic radiologist 1)	0.988 (0.983–0.991)	
	Intra-observer agreement (thoracic radiologist 2)	0.977 (0.968–0.983)	
	Inter-observer agreement (thoracic radiologist 1 vs. thoracic radiologist 2)	0.972 (0.953–0.982)	0.192
	Inter-observer agreement (a deep learning-based model versus two thoracic radiologists)	0.959 (0.945–0.971)	
Chest radiographs without any lung or pleural abnormality	Intra-observer agreement (thoracic radiologist 1)	0.980 (0.962–0.989)	
	Intra-observer agreement (thoracic radiologist 2)	0.998 (0.997–0.999)	
	Inter-observer agreement (thoracic radiologist 1 vs. thoracic radiologist 2)	0.983 (0.968–0.997)	0.868
	Inter-observer agreement (a deep learning-based model versus two thoracic radiologists)	0.982 (0.970–0.991)	
Chest radiographs with pneumothorax	Intra-observer agreement (thoracic radiologist 1)	0.990 (0.981–0.995)	
	Intra-observer agreement (thoracic radiologist 2)	0.992 (0.984–0.996)	
	Inter-observer agreement (thoracic radiologist 1 vs. thoracic radiologist 2)	0.985 (0.976–0.992)	0.534
	Inter-observer agreement (a deep learning-based model versus two thoracic radiologists)	0.982 (0.971–0.990)	
Chest radiographs with pleural effusion	Intra-observer agreement (thoracic radiologist 1)	0.993 (0.987–0.996)	
	Intra-observer agreement (thoracic radiologist 2)	0.969 (0.943–0.984)	
	Inter-observer agreement (thoracic radiologist 1 vs. thoracic radiologist 2)	0.975 (0.959–0.999)	0.066
	Inter-observer agreement (a deep learning-based model versus two thoracic radiologists)	0.949 (0.916–0.982)	
Chest radiographs with consolidation	Intra-observer agreement (thoracic radiologist 1)	0.989 (0.980–0.994)	
	Intra-observer agreement (thoracic radiologist 2)	0.945 (0.899–0.971)	
	Inter-observer agreement (thoracic radiologist 1 vs. thoracic radiologist 2)	0.939 (0.890–0.978)	0.812
	Inter-observer agreement (a deep learning-based model versus two thoracic radiologists)	0.925 (0.878–0.971)	

ICC: intraclass or interclass correlation coefficients. * *p*-values were estimated by 1000 rounds of bootstrapping.

**Table 3 bioengineering-10-01077-t003:** Diagnostic performance for cardiomegaly on chest radiographs for a deep learning-based model and five board-certified radiologists with a reference standard derived using the Dawid–Skene consensus method.

		Sensitivity	Specificity	Positive Predictive Value	Negative Predictive Value	Accuracy
Study sample	Deep learning-based model	96.3% (89.7%–99.2%)	83.3% (73.2%–90.8%)	85.9% (77.0%–92.3%)	95.6% (87.6%–99.1%)	90.0% (84.3%–94.2%)
	Radiologists	97.8% (95.1%–99%)	85.1% (78.8%–89.8%)	87.4% (81.1%–91.8%)	97.4% (93.9%–98.9%)	91.6% (88.1%–94.2%)
	*p*-value *	<0.001	0.005	<0.001	<0.001	<0.001
Chest radiographs without any lung or pleural abnormality	Deep learning-based model	95.5% (77.2%–99.9%)	94.4% (72.7%–99.9%)	95.5% (77.2%–99.9%)	94.4% (72.7%–99.9%)	95% (83.1%–99.4%)
	Radiologists	97.3% (83.3%–99.6%)	97.8% (92%–99.4%)	98.2% (92.6%–99.6%)	96.7% (79.7%–99.5%)	97.5% (91.4%–99.3%)
	*p*-value *	<0.001	0.07	0.021	<0.001	<0.001
Chest radiographs with pneumothorax	Deep learning-based model	100% (80.5%–100%)	82.6% (61.2%–95%)	81% (58.1%–94.6%)	100% (82.4%–100%)	90% (76.3%–97.2%)
	Radiologists	98.8% (92.5%–99.8%)	87% (75.8%–93.4%)	84.8% (69.3%–93.3%)	99% (93.1%–99.9%)	92% (84.8%–96%)
	*p*-value *	<0.001	0.132	0.083	<0.001	0.004
Chest radiographs with pleural effusion	Deep learning-based model	95.2% (76.2%–99.9%)	68.4% (43.4%–87.4%)	76.9% (56.4%–91%)	92.9% (66.1%–99.8%)	82.5% (67.2%–92.7%)
	Radiologists	96.2% (91%–98.4%)	73.7% (57.6%–85.2%)	80.2% (64.1%–90.2%)	94.6% (85.4%–98.1%)	85.5% (76%–91.7%)
	*p*-value *	0.015	0.277	0.071	0.079	0.055
Chest radiographs with consolidation	Deep learning-based model	95.5% (77.2%–99.9%)	88.9% (65.3%–98.6%)	91.3% (72%–98.9%)	94.1% (71.3%–99.9%)	92.5% (79.6%–98.4%)
	Radiologists	99.1% (94%–99.9%)	82.2% (67.2%–91.2%)	87.2% (73.2%–94.4%)	98.7% (90.8%–99.8%)	91.5% (83.2%–95.9%)
	*p*-value *	0.082	0.01	0.003	0.173	0.004

* One-sided non-inferiority test with a cut-off of a *p* value of 0.025.

## Data Availability

The datasets generated or analyzed during the study are available from the corresponding author on reasonable request.

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
