# Peer review of "The Performance of a Deep Learning-Based Automatic Measurement Model for Measuring the Cardiothoracic Ratio on Chest Radiographs"

_bioengineering, 2023, doi:10.3390/bioengineering10091077_

Round 1
Reviewer 1 Report
The manuscript presents a study validating the performance of a commercially available deep learning (DL)-based model for measuring the cardiothoracic ratio (CTR) on chest radiographs. The authors aimed to assess the agreement between the DL model and thoracic radiologists, as well as compare the model's diagnostic performance to that of other radiologists using a probabilistic-based reference standard.
The manuscript is well-written and provides a comprehensive analysis of the performance of a DL-based CTR measurement model, its agreement with thoracic radiologists, and its diagnostic accuracy compared to other board-certified radiologists, making it a thorough and complete study.
It also provides a comprehensive analysis of the study's results, addressing key points related to performance, reference standards, and clinical applicability , although this last one could be further extended.
Minor issue: Figure 4's legend is too long and repeats the information provided in Table 1, Please correct.
Author Response
General Comments for the Editors and Reviewers:
We sincerely thank the Editor-in-Chief, the Associate Editor, and the reviewers for their valuable comments. We have carefully addressed the reviewers’ observations and suggestions and then revised our paper accordingly. Initially, we wish to inform you that certain revisions have been made within the Materials and Methods section with the intention of ensuring the provision of accurate information. Detailed responses to the reviewers’ comments are provided below. The original reviewer’s comments are in black, and the appropriate changes in the annotated manuscript are highlighted in yellow.
Reviewer: 1
The manuscript presents a study validating the performance of a commercially available deep learning (DL)-based model for measuring the cardiothoracic ratio (CTR) on chest radiographs. The authors aimed to assess the agreement between the DL model and thoracic radiologists, as well as compare the model's diagnostic performance to that of other radiologists using a probabilistic-based reference standard.
The manuscript is well-written and provides a comprehensive analysis of the performance of a DL-based CTR measurement model, its agreement with thoracic radiologists, and its diagnostic accuracy compared to other board-certified radiologists, making it a thorough and complete study.
It also provides a comprehensive analysis of the study's results, addressing key points related to performance, reference standards, and clinical applicability , although this last one could be further extended.
Minor issue: Figure 4's legend is too long and repeats the information provided in Table 1, Please correct.
Response: Thank you for your comment. As per your suggestion, we deleted redundant parts in Figure 4 legend.
Reviewer 2 Report
This study presents the evaluation of a commercial DL model while using it in CTR measurements and related diagnostics.
Technically there is no innovation in the paper. In my view the authors should address the following issues.
1. Did the authors test the outcomes using a CTR threshold of 0,5 or they also tested the outcome with other thresholds (e.g. 0,55 or 0,48)? In these cases what was the difference in the model's performance?
2. Why don't the authors construct a t-SNE plot when classifying different pathologies between radiologists and the DL model, as well for the DL model performance itself?
3. Segmentation is important especially in pleural effusion and pneumonia. In these cases CNN approaches or ResNet will definitely enhance outcome. This can tone-up the innovation and the personalised medIcine dimension of such systems.
English is OK.
Author Response
General Comments for the Editors and Reviewers:
We sincerely thank the Editor-in-Chief, the Associate Editor, and the reviewers for their valuable comments. We have carefully addressed the reviewers’ observations and suggestions and then revised our paper accordingly. Initially, we wish to inform you that certain revisions have been made within the Materials and Methods section with the intention of ensuring the provision of accurate information. Detailed responses to the reviewers’ comments are provided below. The original reviewer’s comments are in black, and our the appropriate changes in the annotated manuscript are highlighted in yellow.
Reviewer: 2
This study presents the evaluation of a commercial DL model while using it in CTR measurements and related diagnostics. Technically there is no innovation in the paper. In my view the authors should address the following issues.
- Did the authors test the outcomes using a CTR threshold of 0,5 or they also tested the outcome with other thresholds (e.g. 0,55 or 0,48)? In these cases what was the difference in the model's performance?
Response: Thank you for your helpful comments. As you commented, several prior studies referred to the CTR value of 0.55 for significant cardiomegaly [1-3]. Therefore, we assessed diagnostic performance for cardiomegaly defined as CTR threshold as 0.55 of a deep learning-based model and radiologists. As a result, the deep learning-based model had significantly lower diagnostic measures than those of radiologists in several situations: specificity and positive predictive value in chest radiographs without any lung or pleural abnormality; sensitivity and negative predictive value in chest radiographs with pleural effusion; sensitivity, specificity, positive predictive value, negative predictive value, and accuracy in chest radiographs with consolidation. We suggested this result in the Table S3 of the revised manuscript.
[1] Ajmera P, Kharat A, Gupte T, Pant R, Kulkarni V, Duddalwar V, et al. Observer performance evaluation of the feasibility of a deep learning model to detect cardiomegaly on chest radiographs. Acta Radiol Open 2022;11:20584601221107345. doi: 10.1177/20584601221107345.
[2] Digitalis Investigation Group. The effect of digoxin on mortality and morbidity in patients with heart failure. N Engl J Med 1997;336:525-533
[3] Petrie MC. It cannot be cardiac failure because the heart is not enlarged on the chest X‐ray. Eur J Heart Fail. 2003;5(2):
- Why don't the authors construct a t-SNE plot when classifying different pathologies between radiologists and the DL model, as well for the DL model performance itself?
Response: Thank you for your thoughtful comments. As per your suggestion, we presented plots (Figure S1 of the revised manuscript) that can contain multiple characteristics (cardiothoracic ratio by radiologists and a deep learning-based model, diagnosis of cardiomegaly, and different pathologies), which are similar to the t-SNE.
- Segmentation is important, especially in pleural effusion and pneumonia. In these cases CNN approaches or ResNet will definitely enhance outcome. This can tone-up the innovation and the personalised medIcine dimension of such systems.
Response: Thank you for your valuable comments. The DL model used in this study was built upon a standard UNet architecture and applied self-attention mechanisms to improve discriminative feature representation ability. According to the previous experimental results, it showed reliable segmentation performance in patients with diverse thoracic pathological conditions including pleural effusion [1, 2]. In particular, it exhibited no noticeable difference in segmentation performance when compared to the state-of-the-art CXR segmentation model, XLSor, which utilizes ResNet [1]. However, since we agree with your opinion, we plan to explore more sophisticated algorithms and techniques to improve the overall performance.
[1] Lee, M. S., Kim, Y. S., Kim, M., Usman, M., Byon, S. S., Kim, S. H., ... & Lee, B. D. (2021). Evaluation of the feasibility of explainable computer-aided detection of cardiomegaly on chest radiographs using deep learning. Scientific Reports, 11(1), 16885.
[2] Kim, M., & Lee, B. D. (2021). Automatic lung segmentation on chest X-rays using self-attention deep neural network. Sensors, 21(2), 369.

Reviewer 3 Report
In this study, the authors validated the performance of a commercially available a DL-based CTR measurement model by its agreement with thoracic radiologists, and then they performed reader tests with five radiologists in various thoracic pathologies. The work has a good overall merit. However, the following comments may further help improving the quality of the paper:
1. The presentation of results in the Results section is not clear enough and the results are hard to be understood. It is recommended to include more analysis to the numerical results to make it easy to comprehend.
2. The resolution of the figures 3 and 4 needs to be increased as well as their fonts.
3. The related work is not adequately covered. The introduction section should be extended by more relevant studies in the field.
Minor editing of English language is required.
Author Response
General Comments for the Editors and Reviewers:
We sincerely thank the Editor-in-Chief, the Associate Editor, and the reviewers for their valuable comments. We have carefully addressed the reviewers’ observations and suggestions and then revised our paper accordingly. Initially, we wish to inform you that certain revisions have been made within the Materials and Methods section with the intention of ensuring the provision of accurate information. Detailed responses to the reviewers’ comments are provided below. The original reviewer’s comments are in black, and the appropriate changes in the annotated manuscript are highlighted in yellow.
Reviewer: 3
In this study, the authors validated the performance of a commercially available a DL-based CTR measurement model by its agreement with thoracic radiologists, and then they performed reader tests with five radiologists in various thoracic pathologies. The work has a good overall merit. However, the following comments may further help improving the quality of the paper:
- The presentation of results in the Results section is not clear enough and the results are hard to be understood. It is recommended to include more analysis to the numerical results to make it easy to comprehend.
Response: Thank you for your thoughtful comments. We understand your point. More than anything else, it seems that it may have been confusing about the 95% confidence interval in the agreement analysis. Explanation of these results have been described in the revised manuscript.
- The resolution of the figures 3 and 4 needs to be increased as well as their fonts.
Response: As per your request, we revised the figures 3 and 4.
- The related work is not adequately covered. The introduction section should be extended by more relevant studies in the field.
Response: Thank you for your comments. As you commented, there have been many literatures focusing on the technical aspect of deep learning applied on measuring cardiothoracic ratio on chest radiographs. However, our study is a medical validation study of the deep learning model, and related studies are introduced in the “Introduction” section. Instead, we added other references which suggested necessary points for more robust validation studies in the future in the “Discussion” section. We hope that you understand this point.
Round 2
Reviewer 2 Report
The authors addressed the revisions needed.
English is OK.